# Prompt-to-Leaderboard: Prompt-Adaptive LLM Evaluations

Evan Frick[*][1]  Connor Chen[*][1]  Joseph Tennyson[*][1]  Tianle Li[*][1]  Wei-Lin Chiang[*][1]
Anastasios N. Angelopoulos[*][1]  Ion Stoica[1]

## Abstract

Large language model (LLM) evaluations typically rely on aggregated metrics like accuracy or human preference, averaging across users and prompts. This averaging obscures user- and prompt-specific variations in model performance. To address this, we propose Prompt-to-Leaderboard (P2L), a method that produces leaderboards specific to a prompt or set of prompts. The core idea is to train an LLM taking natural language prompts as input to output a vector of Bradley-Terry coefficients which are then used to predict the human preference vote. The resulting prompt-dependent leaderboards allow for unsupervised task-specific evaluation, optimal routing of queries to models, personalization, and automated evaluation of model strengths and weaknesses. Data from Chatbot Arena suggest that P2L better captures the nuanced landscape of language model performance than the averaged leaderboard. Furthermore, our findings suggest that P2L's ability to produce prompt-specific evaluations follows a power law scaling similar to that observed in LLMs themselves. In January 2025, the router we trained based on this methodology achieved the #1 spot on the Chatbot Arena leaderboard. Our code is available at this GitHub link: https://github.com/lmarena/p2l.

## 1. Introduction

Evaluating the real-world performance of large language models is an unresolved challenge. A growing suite of benchmarks, including MMLU (Hendrycks et al., 2020), MMLU-Pro (Wang et al., 2024), and GPQA (Rein et al., 2023), seek to address the challenge by reporting task-specific performance metrics, such as multiple-choice

question-answering ability. These highly-curated benchmarks focus on domain-specific performance measures but do not capture the general and subjective nature of organic human preferences. Live evaluations, such as Chatbot Arena (Chiang et al., 2024), assess real-world performance by collecting millions of organic human preferences from users who visit the site and vote between pairs of model responses. These pairwise comparisons are aggregated using Bradley-Terry (BT) regression (Bradley and Terry, 1952) to form a leaderboard over LLMs. This leaderboard averages over many users and prompts, only providing a coarse understanding of performance.

For example, if we want to identify the best model for SQL queries, the overall Chatbot Arena leaderboard may not be useful since SQL queries make up only 0.6% of organic submissions and thus have little influence in the ranking. A natural solution is to stratify the data and run a separate BT regression for SQL queries. However, collecting the 3,000-5,000 SQL votes needed for a stable ranking would require around a million total votes—taking months to collect. Finer-grained categories, for example SQL table joins, would demand even more data, making stratified regression impractical and slow. And the finest-grained analyses—for example, producing leaderboards for a *specific* prompt or use-case—are rendered impossible.

This manuscript proposes a solution to this problem via a method called Prompt-to-Leaderboard (P2L). P2L takes a prompt as input and outputs a leaderboard quantifying LLM performance *on that specific prompt*. Thus, P2L can be used to assess which models are best for a specific use-case, as opposed to on average. Per-prompt leaderboards can also be aggregated over a group of prompts to form personalized leaderboards, showing which model is best for an individual or enterprise based on their prompt history.

The system works by training a P2L model, which is an LLM trained on human preference feedback to output a Bradley-Terry (BT) coefficient for every model in question; see Section 2.1. Because P2L characterizes the prompt-conditional win rate of any two models, it enables several downstream applications. These include optimally routing prompts to LLMs (Section 2.1.2), personalized evaluations based on a user's prompt history (Section 2.1.1), automated

---

[*]Equal contribution [1]University of California, Berkeley. Correspondence to: Evan Frick <evanfrick@berkeley.edu>.

*Proceedings of the $42^{nd}$ International Conference on Machine Learning*, Vancouver, Canada. PMLR 267, 2025. Copyright 2025 by the author(s).

strength and weakness analysis of models (Section 3.4), and more. Thus, we view P2L as a general-purpose tool for highly granular evaluations extracted from large corpuses of preference data. As a demonstration of P2L's utility, we tested our prompt routing strategy on Chatbot Arena between the dates 01/19/2025—01/27/2025, and it achieved the #1 spot with a score increase of 25 points over the previous top model, `Gemini-exp-1206` (see "P2L router performance" in Figure 1).

More broadly, P2L is a subclass of a more general methodology we call Prompt-to-Regression (P2R) for training LLMs to output coefficients of parametric statistical regressions (see Section 2.2). A canonical example that we will develop throughout this paper is a model taking prompts as input and outputting Bradley-Terry coefficients, as mentioned earlier. However, the method also accommodates other feedback models (ties, real values, etc.) via other parametric models. We describe this method and derive the optimal routing strategy in Section 2. We show experiments and other applications in Section 3.

## 2. Methods

We describe the P2L method formally, beginning with notation. Consider $M$ different LLMs which are presented to humans pairwise—model $A$ on the left, and model $B$ on the right, where $A$ and $B$ are randomly sampled without replacement from $[M] = \{1, \ldots, M\}$. If the human votes for model $A$, we set $Y = 0$, and if they vote for model $B$, we set $Y = 1$. Furthermore, we let $X$ represent a 'two-hot' encoding of the model pair, i.e., a vector of length $M$ with zeros everywhere except $+1$ in the index $B$ and $-1$ in the index $A$. We model our data-generating process as a tuple $(X, Y, Z)$ of two-hot encodings, votes, and prompts $Z \in \mathcal{Z}$ sampled from a joint distribution $P$, where $\mathcal{Z}$ denotes the space of natural-language prompts. Also, let $\Theta$ denote a space of functions mapping prompts to leaderboards, i.e., $\theta \in \Theta$ is a function from $\mathcal{Z} \to \mathbb{R}^M$, and $\theta(z)_i$ represents the leaderboard score of model $i \in [M]$ given prompt $z$. Finally, let $\ell$ denote the binary cross-entropy loss and $\sigma$ denote the sigmoid function.

### 2.1. Core method

Conceptually, our method works as follows. We model the vote conditionally on the prompt and model pair as following a Bradley-Terry (BT) model (Bradley and Terry, 1952):

$$\mathbb{P}(Y = 1 \mid X = x, Z = z) = \sigma(x^\top \theta^*(z)),$$

for some (unknown) $\theta^* : \mathcal{Z} \to \mathbb{R}^M$. The goal is to approximate $\theta^*$ from data.

For any prompt $z \in \mathcal{Z}$, $\theta^*(z)$ represents a leaderboard.

Each model $m \in [M]$ has a coefficient $\theta^*(z)_m$, and the higher this coefficient is, the more likely model $m$ beats any other model on the prompt $z$. For different prompts, the leaderboard will be different, capturing the idea that different models are better on different prompts. Our target, $\theta^*$, is precisely the function that takes prompts and outputs leaderboards—hence the name, *prompt-to-leaderboard* (P2L).

P2L is a strict generalization of marginal BT regression. In marginal BT regression, we simply omit the dependence of the leaderboard on the prompt, and give the best leaderboard on average ("marginally"). That is, choosing $\Theta$ to be the class of *constant* functions $\theta(z) \equiv \theta \in \mathbb{R}^M$ exactly recovers marginal BT regression.

However, P2L can be substantially more powerful than marginal BT regression due to heterogeneity in the prompt-conditional performance of different language models. That is, we should leverage language models to extract information on model performance from the prompt. In particular, our work takes $\Theta$ to be a space of reward models mapping prompts to vectors. Given a training dataset $D^{\text{train}} = \{(X_i, Y_i, Z_i)\}_{i=1}^N$, we find the empirical risk minimizer,

$$\hat{\theta} = \underset{\theta \in \Theta}{\arg\min} \frac{1}{N} \sum_{i=1}^N \ell(\sigma(X_i^\top \theta(Z_i)), Y_i). \qquad (1)$$

Then, as before, we can extract the estimated win rate between any two models as

$$\widehat{\mathbb{P}}(Y = 1 \mid X = x, Z = z) = \sigma(x^\top \hat{\theta}(z)).$$

Lastly, we note that this strategy of training LLMs to output coefficients of parametric statistical models will be generalized in Section 2.2. The resulting prompt-dependent models have both high predictive power and a useful statistical interpretation, which is critical to the aforementioned routing and personalization techniques.

#### 2.1.1. AGGREGATING LEADERBOARDS

Many practical scenarios require a leaderboard for a distribution over prompts, not just one. For example, a user may want to know which model is best for them based on their chat history, or an enterprise may want to know which model is best for their use-case. In other words, given a distribution over prompts $Q$, we want to ensemble all $\theta^*(z)$ for $z \in \mathcal{Z}$ to form a leaderboard over $Q$. In the case of a finite chat history, we can consider $Q$ to be the discrete uniform distribution over the observed historical prompts.

By the Tower property, we can decompose the win rate as

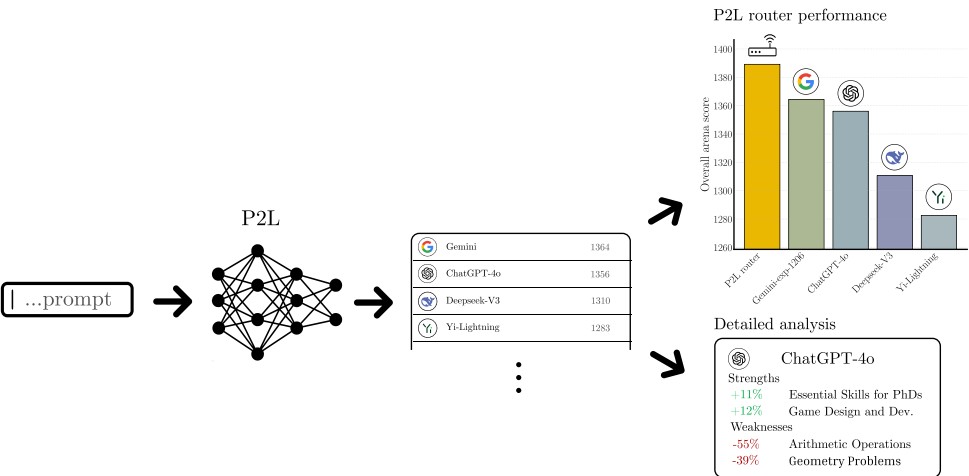

Figure 1: **Pipeline of P2L.** P2L takes a prompt or a set of prompts and outputs an $M$-dimensional vector that we call a leaderboard. Once we have a leaderboard, we can build better data products, like routers and automatic analyses (see right).

$$\mathbb{E}_{Z\sim Q, Y\sim \text{Bern}(\sigma(X^\top\theta^*(Z)))}[Y \mid X = x]$$
$$= \int_{z\in\mathcal{Z}} \sigma\left(x^\top\theta^*(z)\right) dQ(z).$$

The win rate above no longer follows a simple logistic model, but we can fit another logistic model to match it:

$$\tilde{\theta}(Q) = \underset{\theta\in\Theta}{\text{argmin}}\, \mathbb{E}_{\substack{X\sim P_X, Z\sim Q, \\ Y\sim \text{Bern}(\sigma(X^\top\theta^*(Z)))}} \left[\ell\left(\sigma(X^\top\theta), Y\right)\right]. \tag{2}$$

The idea is that, because we know $\mathbb{P}(Y = 1 \mid X = x, Z = z) = \sigma(x\top\theta^*(z))$ for all $x$ and $z$, we can simulate the data-generating process. This allows us to construct a synthetic dataset and fit a Bradley-Terry model to it. If $\theta^*$ exists, this technique is perfect, in that it recovers the exact same BT coefficients that we would have obtained by observing an infinite population of prompts from $Q$. In Appendix B.1, we explore an alternative leaderboard aggregation strategy by taking a weighted average of the leaderboards. Note also that we use $\theta^*$, with the understanding that in practice we will use the plug-in estimate based on $\hat{\theta}$, and the resulting rule will be approximate.

We can make this strategy more efficient by leveraging the linearity of the binary cross-entropy loss. Namely,

$$\mathbb{E}_{X\sim P_X, Z\sim Q, Y\sim \text{Bern}(\sigma(X^\top\theta^*(Z)))} \left[\ell\left(\sigma(X^\top\theta), Y\right)\right]$$
$$= \mathbb{E}_{X\sim P_X, Z\sim Q} \left[\mathbb{E}_{Y\sim \text{Bern}(\sigma(X^\top\theta^*(Z)))}\right.$$
$$\left.\left[\ell\left(\sigma(X^\top\theta), Y\right) \mid X, Z\right]\right]$$
$$= \mathbb{E}_{X\sim P_X, Z\sim Q} \left[\ell\left(\sigma(X^\top\theta),\right.\right.$$
$$\left.\left. \mathbb{E}_{Y\sim \text{Bern}(\sigma(X^\top\theta^*(Z)))}\left[Y|X, Z\right]\right)\right]$$
$$= \mathbb{E}_{X\sim P_X, Z\sim Q} \left[\ell\left(\sigma(X^\top\theta), \sigma(X^\top\theta^*(Z))\right)\right].$$

Thus, we can bypass the need for sampling to simulate $Y$. In other words, (2) is equivalent to

$$\tilde{\theta}(Q) = \underset{\theta\in\Theta}{\text{argmin}}\, \mathbb{E}_{X\sim P_X, Z\sim Q} \left[\ell\left(\sigma(X^\top\theta), \sigma(X^\top\theta^*(Z))\right)\right]. \tag{3}$$

This last expression is simple to compute for discrete distributions $Q$, leading to an efficient algorithm.

### 2.1.2. OPTIMAL ROUTING

Next, we will derive the optimal router based on P2L. We will derive the exact optimal router based on $\theta^*$ and approximate it in practice by $\hat{\theta}$. Let us assume, for the sake of simplicity, that for each model $m \in \{1, \ldots, M\}$, there is a known and fixed cost of inference, $c = (c_1, \ldots, c_M)$. We seek to create a router that maximizes performance while remaining below a constraint on the average cost, $C$. We express the router as a policy, $\pi : \mathcal{Z} \to \Delta^M$, which takes a prompt as input and outputs a distribution over models; we seek to estimate the optimal policy, $\pi^*$. We will also consider a distribution of opponent models, $q \in \Delta^M$, to act as a baseline for comparison. For instance, we can pick $q$ to be a point-mass on the single best model, or to be uniform over all $[M]$ models.

One possible interpretation of an "optimal" router is the one that maximizes the win rate against $q$ subject to the cost constraint; that is, for almost every $z$, this interpretation of $\pi^*(z)$ solves the following optimization problem:

$$\underset{\tilde{\pi}\in\Delta^M}{\text{maximize}}\quad \mathbb{P}_{A\sim q, B\sim\tilde{\pi}, Y\sim\text{Bern}(\sigma(\theta^*(z)_B - \theta^*(z)_A))}$$
$$(Y = 1 \mid Z = z) \tag{4}$$
$$\text{subject to}\quad \mathbb{E}_{B\sim\tilde{\pi}}[c_B] \leq C$$

In other words, the optimal router should maximize the

average win rate against the opponent distribution $q$.

An alternative definition of the optimal router is the one that has the highest Bradley-Terry coefficient. This version of the optimal policy has $\pi^*(z)$ equal (almost surely) to the solution to the following optimization problem:

$$
\begin{aligned}
&\underset{\tilde{\pi} \in \Delta^M}{\text{maximize}} \quad \underset{\theta \in \mathbb{R}}{\arg\min} \, \mathbb{E}_{\substack{B \sim \tilde{\pi}, A \sim q, \\ Y' \sim \text{Bern}(\sigma(\theta^*(z)_B - \theta^*(z)_A))}} \\
&\qquad \Big[ \ell(\sigma(\theta - \theta^*(z)_A), Y') \mid Z = z \Big] \\
&\text{subject to} \quad \mathbb{E}_{B \sim \tilde{\pi}}[c_B] \leq C
\end{aligned} \tag{5}
$$

That is, considering the optimal router as a separate model, it should achieve the highest possible spot in the leaderboard subject to the cost constraint.

Surprisingly, although the optimization problems in (4) and (5) look different, their optimal solution is the same under the Bradley-Terry model. The solution is given in Theorem 1. The resulting problem has a linear objective and a linear constraint, and can be solved with any standard solver. If the dominant model is below the cost of $C$, the policy will deterministically select that model (i.e., it will place probability 1 on sampling that model). Otherwise, it will hedge its bets and randomize over multiple models.

**Theorem 1** (Optimal prompt-dependent routing). *Assume that for every prompt $z$, the Bradley-Terry model holds with coefficients $\theta^*(z)$. Then, the optimization problems in (4) and (5) are both equivalent to the following problem:*

$$
\begin{aligned}
&\underset{\tilde{\pi} \in \mathbb{R}^M}{\text{minimize}} \quad -\tilde{\pi}^\top \mathbf{W}^* q \\
&\text{subject to} \quad \tilde{\pi}^\top c \leq C, \\
&\qquad\qquad \mathbf{0}_M \preceq \tilde{\pi} \preceq \mathbf{1}_M \\
&\qquad\qquad \tilde{\pi}^\top \mathbf{1}_M = 1,
\end{aligned} \tag{6}
$$

*where $\mathbf{W}^*$ represents the population win matrix, with entries $\mathbf{W}^*_{ba} = \sigma(\theta^*(z)_b - \theta^*(z)_a)$.*

The proof is given in Appendix A. It is important to note that deviations from the Bradley-Terry model—for example, any non-transitivity—will break this relationship.

Another benefit of this approach is that we are able to estimate the *value* of the objective function of (5) via a standard root finder (Brent, 1973), which means we can estimate the router's position on the leaderboard before deploying it. We give this procedure in Algorithm 1. It is justified by (9) in the proof of Theorem 1.

---

**Algorithm 1** Optimal routing with BT estimate

**Input:** $q$; $\mathbf{W}^*$; $\theta^*(z)_j$; $c$; $C$
1: Solve the LP:
$$
\tilde{\pi}^* = \underset{\tilde{\pi} \in \Delta^M, \, \tilde{\pi}^\top c \leq C}{\arg\max} \, \tilde{\pi}^\top W^* q
$$

2: Compute $R^* = \tilde{\pi}^{*\top} W^* q$
3: Solve for $\theta'^*$ by finding the root of the following implicit equation:
$$
\sum_a q_a \, \sigma(\theta - \theta^*(z)_a) = R^*
$$

**Output:** Optimal router $\tilde{\pi}^*$, estimate of router's BT coefficient $\theta'^*$

---

### 2.2. Prompt-to-Regression

Here, we give extensions of P2L beyond pairwise preference feedback. This is useful because, in Chatbot Arena, the voting options are not just "A is better" and "B is better"; they also include "Tie" and "Tie (both bad)". Thus, a P2L model that takes into account all this additional data may learn faster and also learn interesting signals about which prompts are hard and cause models to exhibit different behaviors or failures. Fortunately, our toolkit generalizes to the case where $X$ is no longer a two-hot encoding and $Y$ is no longer binary. In fact, our strategy encompasses any parametric statistical model relating $X$ and $Y$ conditionally on $Z$, regardless of the space in which they live. We call this more general class of models *prompt-to-regression* models.

More formally, let us model the distribution of $Y$ by saying that for all putative values $y$,

$$
p_{Y=y|Z=z, X=x}(y) = g_{\theta^*(z)}(y; x), \tag{7}
$$

for some (unknown) vector of parameters $\theta^*(z)$. Then, we fit $\hat{\theta}(z)$ by running maximum-likelihood estimation, i.e., maximizing $\prod_{i=1}^n g_{\theta(Z_i)}(Y_i; X_i) p_X(X_i)$. As a familiar example, we can set $g_{\theta^*(z)}$ to a BT model relating $X$ and $Y$:

$$
g_{\theta(z)}(y; x) = \begin{cases} \sigma(x^\top \theta^*(z)) & y = 1, \\ 1 - \sigma(x^\top \theta^*(z)) & y = 0. \end{cases}
$$

Note that the formulation of (7), $Y$ and $X$ can be arbitrary, so long as we model their conditional relationship via $g_{\theta(z)}$. Thus, the framework can admit real-valued feedback $Y$ via ordinary least squares, count feedback via Poisson regression, and so on.

As one example, we will consider incorporating ties via a Rao-Kupper (Rao and Kupper, 1967) model. Let $X$ be a

two-hot encoding, $Y \in \{\mathsf{A}, \mathsf{B}, \mathsf{tie}\}$, and

$$
g_{\theta^*(z)}(y; x) = \begin{cases} \sigma((x, -1)^\top \theta^*(z)) & y = \mathsf{B}, \\ \sigma((-x, -1)^\top \theta^*(z)) & y = \mathsf{A}, \\ 1 - \sigma((-x, -1)^\top \theta^*(z)) \\ \quad - \sigma((x, -1)^\top \theta^*(z)) & y = \mathsf{tie}. \end{cases}
$$

In this technique, $\theta^*(z)$ is an $(M+1)$-dimensional vector, the last entry of which encodes a tie coefficient. The larger this prompt-dependent tie coefficient, the more likely the two models are to tie. Meanwhile, the first $M$ entries, $\hat{\theta}(z)_{1:M}$, comprise the leaderboard.

Finally, we consider how to handle the "Tie (both bad)" category. For this, we developed a non-standard statistical model which we call the *grounded* Rao-Kupper model. In this model, if both model coefficients are small, it increases the probability of "Tie (both bad)". Inspired by the Plackett-Luce model (Plackett, 1975; Luce, 1959), we imagine the existence of a fictitious "bad" model with a coefficient of zero, and use this as a grounding point for the model coefficients.

Let $Y \in \{\mathsf{A}, \mathsf{B}, \mathsf{tie}, \mathsf{bad}\}$, and for the sake of notational convenience, let $\theta^*(z) = (\beta^*(z), \lambda^*(z))$ where $\beta^*(z) \in \mathbb{R}^M$ and $\lambda^*(z) \in \mathbb{R}_{\geq 1}\}$. For notational convenience, we define $\varphi^*(z)_i := \exp(\beta^*(z)_i)$. The grounded Rao-Kupper model is defined as:

$$
g_{\theta^*(z)}(y; x) = \begin{cases} \frac{\varphi^*(z)_A}{\varphi^*(z)_A + \lambda^*(z)\varphi^*(z)_B + 1} & y = \mathsf{A} \\ \frac{\varphi^*(z)_B}{\varphi^*(z)_B + \lambda^*(z)\varphi^*(z)_A + 1} & y = \mathsf{B} \\ \frac{1}{1 + \varphi^*(z)_A + \varphi^*(z)_B} & y = \mathsf{bad} \\ 1 - \frac{\varphi^*(z)_A}{\varphi^*(z)_A + \lambda^*(z)\varphi^*(z)_B + 1} \\ \quad - \frac{\varphi^*(z)_B}{\varphi^*(z)_B + \lambda^*(z)\varphi^*(z)_A + 1} \\ \quad - \frac{1}{1 + \varphi^*(z)_A + \varphi^*(z)_B} & y = \mathsf{tie}. \end{cases}
$$

$$(8)$$

This model allows us to make efficient use of all data collected on Chatbot Arena by incorporating all votes. It also has the additional advantage that models with higher coefficients have a lower probability of being labeled "Tie (both bad)". Thus, the raw coefficient value of a model speaks to its absolute quality, as opposed to its comparative quality against other LLMs as in the BT model.

## 3. Experiments

This section contains a suite of experiments that validate the P2L method and demonstrate its utility. In Section 3.2, we show that P2L leads to gains in human preference prediction that scale with model size and data. In Section 3.2, we show direct predictive performance on pairwise human preferences, as well as scaling behavior with data size and parameter count. In Section 3.3, we show P2L allows for

optimal cost-efficient routing via the algorithm developed previously in Section 2.1.2. In Section 3.4, we use P2L to automatically identify strengths and weaknesses for different models. In Section 3.5, we explore our aggregation technique against ground truth categories leaderboards, and observe data scaling trends. Finally, in Section 3.6, we show that the P2L has reasonable performance on out-of-distribution data.

### 3.1. Training setup

To train a P2L model, we follow this three-step procedure: **(1)** Begin with a pre-trained, instruction-tuned LLM. **(2)** Remove the existing language model head and replace it with a randomly initialized *coefficient head*. In the BT case, the coefficient head is a linear layer producing $M$ outputs, one per model. **(3)** Train the model by running stochastic gradient descent on all parameters to minimize the negative log-likelihood: $\mathcal{L}(\theta) = -\sum_{i=1}^{n} \log\left(g_{\theta(Z_i)}(Y_i; X_i)\right).$ The result of this procedure is the trained model $\hat{\theta} = \arg\min_{\theta \in \Theta} \mathcal{L}(\theta)$, which is a direct generalization of (1).

We train on up to $n = 1.5$ million crowdsourced human preference pairs from Chatbot Arena, containing $M = 130$ unique models. Note that we find minimal left/right positional bias from voters. We always train for 1 epoch. In order to study the scaling laws of P2L as a function of model size, we used the following models as the initializations: `SmolLM2-{135, 360}M-Instruct` and `Qwen2.5-{0.5, 1.5, 3, 7}B-Instruct` (Allal et al., 2024; Team, 2024). We refer to our post-trained versions of these models as `P2L-{135,360}M` and `P2L-{0.5,1.5,3,7}B`, respectively.

### 3.2. Feedback prediction

We begin by evaluating P2L on its ability to predict human feedback on a prompt-by-prompt basis. In other words, given two models and a prompt, we ask how effectively P2L can predict which model will win on that prompt. These experiments measure the ability of P2L to accurately assess relative model quality on a prompt-by-prompt basis.

In this section, we evaluate the ability of P2L to predict human preferences on Chatbot Arena. We construct a holdout validation set containing 41,507 annotated pairwise comparisons across 34 well-used models. We then measure the negative log-likelihood (validation loss) on this dataset; a lower validation loss indicates better preference prediction performance.

Figure 2 shows the results of our procedure against two baselines. First, we include the constant predictor that gives an equal probability of all preference outcomes; this is an

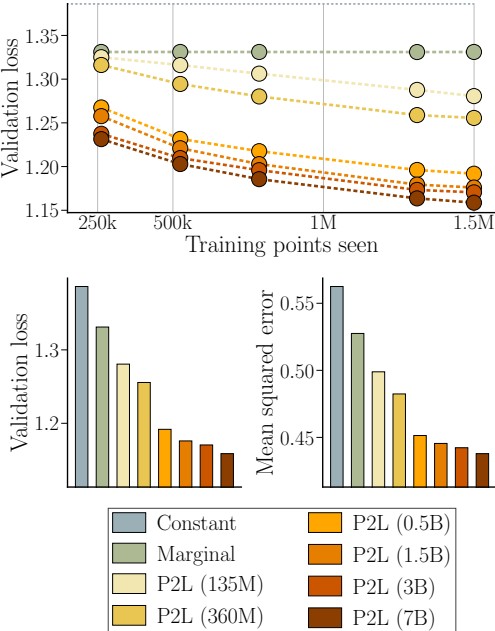

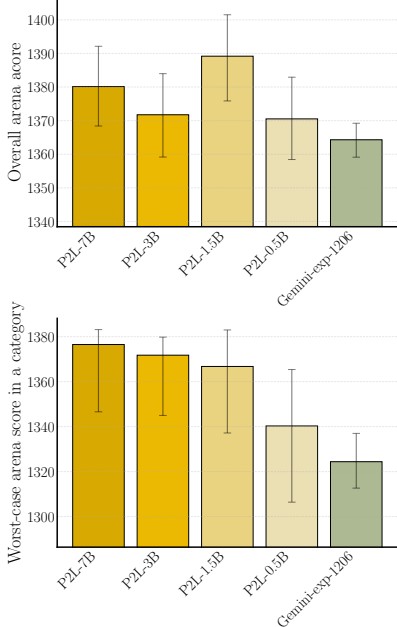

Figure 2: **Loss metrics.** The line plot shows the validation loss as a function of the number of data points seen during training. The P2L models all substantially outperform the baselines, and performance scales with dataset and model size. The bar plots show the validation loss and mean squared error of the models trained on all 1.5M training points. A table for accuracy can be found in Appendix E.2.

Figure 3: **P2L router performance** on Chatbot Arena. The left barplot shows the overall score of the router after it was deployed prospectively on Chatbot Arena. The right barplot shows the worst-case category score on Chatbot Arena. Overall, larger models lead to higher Arena scores, i.e., better routers. The exception is `P2L-1.5B`, which has a large bump in overall performance. However, the confidence intervals indicate that this bump is explainable by statistical variations in its BT coefficient estimate.

extremely weak baseline akin to flipping a coin to decide the winner. Second, we include the average ("marginal") leaderboard. For P2L, we show a ladder of increasing model and dataset sizes. The more data is used to train P2L, the better the preference predictions become. Notably, the gap between the best P2L leaderboard and the marginal model is several times the gap between the marginal leaderboard and the constant predictor. This indicates that by capturing the prompt-dependent differences in model performance, P2L is able to produce much better predictions of human preference.

### 3.3. Optimal routing

Next, we evaluate the performance of the optimal router based on P2L as derived in Section 2.1.2. Our evaluations are based on prospective deployments of our router to Chatbot Arena. We treat the router as a separate model. For all deployments of our routers, we collect blind pairwise comparisons against all active public models hosted on Chatbot Arena in a process identical to how standard models are added to the Chatbot Arena Leaderboard.

We deployed the grounded Rao-Kupper versions of `P2L-0.5B`, `P2L-1.5B`, `P2L-3B`, and `P2L-7B` onto

Chatbot Arena, crowdsourcing a total of 8,616 pairwise comparisons between P2L models and public models hosted on Chatbot Arena. The P2L models routed between 34 models, including top models such as `Gemini-exp-1206`, `o1-2024-12-17`, and `ChatGPT-4o-20241120` as well as other models. (See Appendix E.4 for a full model list.)

Because there is no cost-constraint, the P2L router always picks the highest-ranked model conditionally on the prompt, i.e., the highest entry in $\hat{\theta}(z)$. Marginally, the strongest singular candidate model in the P2L router was `Gemini-exp-1206`, with a score of 1364.

As shown in the top plot in Figure 3, all P2L routers, regardless of parameter count, outperformed `Gemini-exp-1206`. The best model, `P2L-1.5B`, reached #1 on Chatbot Arena during our testing period with a score of 1389. This shows the utility of P2L: differences in model performance on a prompt-by-prompt basis allow P2L to outperform all individual LLMs.

Next, we discuss scaling performance with respect to the Arena score of the router. We see a general trend in Fig-

ure 3 that bigger models do better overall. The exception is `P2L-1.5B`, whose performance was unexplainably strong; otherwise, the trend holds. We also tested other metrics, such as worst-case performance (bottom of Figure 3). The worst-case performance of P2L scales with parameter count as expected, and is uniformly much better than that of the marginal leaderboard.

We also observe that the gap between the P2L routers and static models is large. The P2L routers are able to avoid per-prompt model weaknesses and route elsewhere. In fact, the gap between the best P2L router and the best non-routed static model in the overall comparison was 25 points, while this gap grew to 51 points in the minimum category performance case. Appendix Figure 7 shows `P2L-7B`'s routing distribution conditioned on each Chatbot Arena category. Notably, we see relatively diverse routing patterns, even within a single category. We also observe intuitive behavior patterns, such that heavily routing to `o1-2024-12-17` for math prompts and `Gemini-exp-1206` for creative prompts.

### 3.3.1. COST-OPTIMAL ROUTING

We show results of the optimal routing procedure detailed in Theorem 1 with a `P2L-7B` model on Chatbot Arena. Here, we use P2L to route between `o1-mini`, `gpt-4o-2025-05-13`, `claude-3-5-sonnet-20240620`, `gemini-1.5-pro-001`, `mistral-large-2407`, `claude-3-5-haiku-20241022`, and `gemini-1.5-flash-001` and with budgets of $\{0.00218, 0.0044, 0.00675, 0.00945, 0.0123, \infty\}$. To get reasonable cost estimates, we calculate the expected cost per query with $c_i = O_i * \mathbb{E}[T_i]$ for all models $i \in [M]$, where $O_i$ is the output cost per token of model $i$, and $T_i$ is a random variable representing the number of tokens in a response from model $i$. We estimate $\mathbb{E}[T_i]$ as the response token length mean overall responses from model $i$ in Chatbot Arena. Additionally, we estimate $q$ in Theorem 1 according to the Chatbot Arena model sampling distribution. We find the P2L router performs well, with Pareto frontier Arena score versus cost. Furthermore, on the right plot in Figure 4 we find the P2L router continues to show dominant performance in Chatbot Arena's creative category despite large shifts in individual model performances.

### 3.4. Testing for regression and strength/weakness analysis

An important question when developing models is to understand their category-level performance, along with strengths and weaknesses. Imagine, for example, a business seeking to upgrade their workflow to a cheaper or newer (and pre-

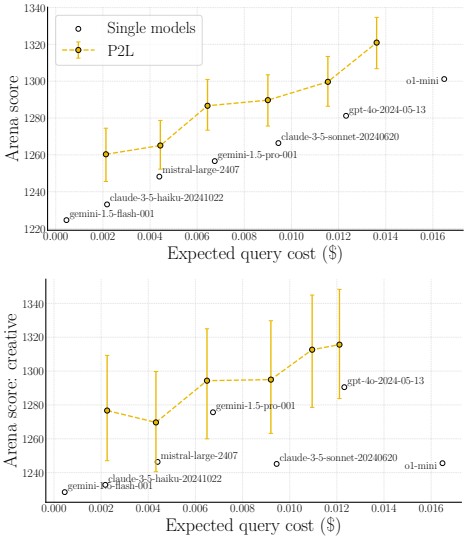

Figure 4: **Arena score versus cost**. Both plots show routing performance as a function of average cost. The left plot shows the averaged performance across all categories, and the right plot shows the performance in the creative writing category. The black open circles give the raw performance and cost of the models used by the router. Each gold dot represents the Arena score of the `P2L-7B` router as a function of the cost constraint in (6). The plots show that the P2L router dominates and substantially improves the cost-performance Pareto frontier. All confidence intervals are 95%.

sumably more advanced) model. In such a business, testing for regression of the model to a worse performance may be important. For example, they might ask the question: if I switch from `GPT-4o` to `GPT-4o-mini`, can I do so safely, and will my performance get worse on my customers?

This is a challenging question to answer because it requires knowledge of the enterprise's customer distribution which may require lengthy instrumentation and data collection procedures. However, P2L provides a partial solution to this problem. Given a large unlabeled dataset of prompts (e.g., customer use-cases), we seek to: (1) Categorize these prompts automatically using an LLM. (2) Produce a preference leaderboard within each category, and (3) On a per-model basis, analyze for which categories it is weak and strong (relative to itself or its competition).

For this, we can use a standard hierarchical clustering approach. Assume access to a multilevel hierarchical categorization of prompts (this can be obtained from an LLM). That is, we have a function `categorize` that takes in a prompt $z$ and an integer level $l$ and outputs a category in $\{1, \ldots, k_l\}$, for some integer $k_l$. Given a set of prompts, $\mathcal{Z}^{\text{category}}$, we can compute a per-category leaderboard using $\tilde{\theta}(\text{unif}(\mathcal{Z}))$

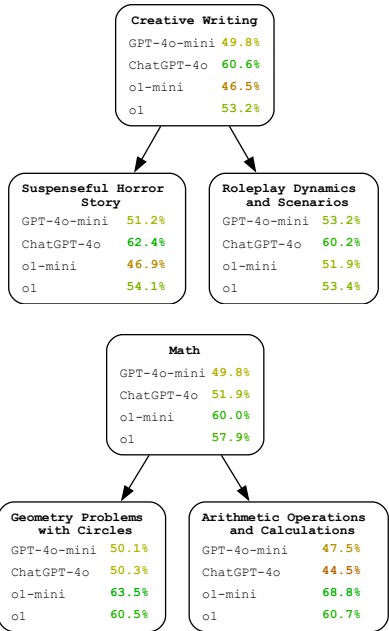

Figure 5: **Regression test.** We show the strengths of different OpenAI models on various topic clusters based on their win rate against `GPT-4o-2024-05-13` as predicted by `P2L-7B`. For each category, we show the probability a given model wins against `GPT-4o-2024-05-13` under the BT model. The results show strong category-specific variability in performance; for example, `o1-mini` is substantially better than `GPT-4o-2024-05-13` in "Arithmetic Operations and Calculations" but substantially worse when asked to write a "Suspenseful Horror Story".

as in (3). Note that the finest-grained categories may have very little data, motivating the need for P2L.

Figure 5 shows an example analysis of five different OpenAI models. The clustering method used is detailed in Appendix Section E.1. Here, the percentages are calculated as the win rate against `GPT-4o-2024-05-13` under the BT model. According to `P2L-7B`, OpenAI models' performance varies across different categories and topic clusters. While `o1` might be a better model on average, it is essentially the same compared to `GPT-4o-mini` on certain creativity tasks, notably the former is 100x more expensive than the latter. In math-flavored tasks, the gap widens significantly. These intuitive results demonstrate the reliability and effectiveness of P2L. See Figures 8 and 9 for similar and more detailed plots on Llama 3 fine-tunes. We also include a variant of our regression analysis under the grounded RK model from (8); this provides guidance as to the absolute reliability of the model, not just preference over alternative models; see Figure 10.

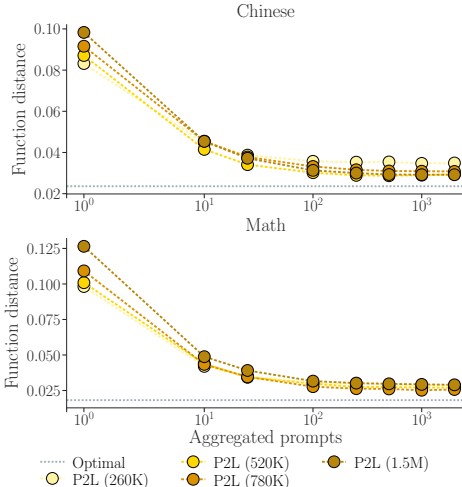

Figure 6: **Aggregation scaling.** The L1 distance between the aggregated leaderboard and the marginal BT regression as a function of the number of randomly sampled and aggregated datapoints in two categories: Chinese (left) and Math (right). The L1 distance plateaus at the optimal performance, which is around 0.025. A nonzero optimal distance is expected as the empirical BT coefficients are derived from a finite validation sample, and so these coefficients have their own irreducible statistical error. Thus, the P2L estimate converges to a near-optimal solution with increased data.

### 3.5. Aggregation scaling

Given a distribution of prompts, we aim to evaluate how P2L behaves using the aggregation technique described in 2.1.1. Specifically, we analyze how P2L's aggregated leaderboards compare to ground truth category leaderboards as well as how this relationship scales with data. First, we calculate ground truth leaderboards over a large category from the validation set with marginal regression. We then aggregate P2L over increasing subsets of this category's prompts. Lastly, we plot the L1 function distance between the aggregated leaderboard's predicted probabilities and the ground truth leaderboard's predicted probabilities as subset size increases. Since both the train and validation set are drawn from the same distribution, we denote the optimal value to be the L1 function distance between the ground truth category leaderboard and the category leaderboard derived from marginal regression on the train set.

In contrast to marginal regression, which requires thousands of prompts for a stable leaderboard, P2L converges near this optimal value within 100-250 prompts (Figure 6). Here, we see P2L's potential to create accurate aggregated leaderboards efficiently, while also reinforcing the validity of its per prompt outputs. Furthermore, as we scale the amount of training data seen, P2L's predictions over singular prompts differ more drastically from category leaderboards while

still converging with more prompts (Figure 6). A clear scaling law ensues, as increased data allows P2L to make more distinguished individual leaderboards while still maintaining its aggregation ability at the category level.

### 3.6. Performance on out-of-distribution prompts

To assess how P2L generalizes to unseen prompts, we evaluate it on LiveBench (White et al., 2024), a verifiable, contamination-free benchmark with 1,000 questions covering diverse categories (e.g., math, coding, reasoning). Unlike Chatbot Arena, it utilizes objective success metrics. We restrict our evaluation to a smaller pool of models. Among these models, P2L selects its candidate models for each question based on the predicted prompt-specific performance and then uses the output of the chosen model as the final answer. Table 1 shows that `P2L-7B` surpasses the static oracle baseline among the model subset, achieving an overall LiveBench score of 59.275. Even far smaller versions (e.g., 1.5B) match or exceed top static models. This means P2L, having never seen ground truth labels or model responses, performs as well or better than running all models on LiveBench, scoring them using the benchmark's ground truth labels, and selecting the best model *after the fact*. This demonstrates that preference-trained routing generalizes well to an out-of-distribution, ground-truth benchmark.

Moreover, we consider the cost-constrained routing case. To examine this trade-off, we apply Prompt2Leaderboard to LiveBench at various cost thresholds (e.g., $2, $5, $10, $15 per million tokens) using the cost-optimal routing method discussed in Section 3.3.1. Figure 11 (in the appendix) shows that, in all budgets tested, the P2L cost-aware router consistently scores higher or comparable LiveBench scores to the best-performing model within that specific cost threshold. These gains are most pronounced when the budget permits occasional routing to a more expensive (and often stronger) model for prompts that particularly benefit from it.

## 4. Discussion and Related Work

This work develops fundamental tools for granular and query-specific evaluations in all evaluation tasks. Although our experiments are largely based on Chatbot Arena, this is not the only evaluation that could benefit from P2L. As discussed in Section 2, any feedback signal can be accommodated. Thus, our techniques would equally work well for other evaluations (Hendrycks et al., 2020; Zellers et al., 2019; Cobbe et al., 2021; Srivastava et al., 2023; Zhong et al., 2023; Chen et al., 2021; Lin et al., 2023; Liang et al., 2022) as well as cost and latency prediction.

**Modeling human preference.** During Reinforcement Learning from Human Feedback (RLHF), a reward model is often trained as a proxy to human preference. Similar to P2L, reward model training may use a contrastive pairwise or $K$-wise loss, for example using the BT model (Christiano et al., 2023; Bai et al., 2022; Ouyang et al., 2022; Zhu et al., 2023). However, reward models are agnostic to model identity, requiring a prompt and response to return a single score for the response. P2L, which is aware of model identities, instead seeks to output expected model response quality, conditioned on input prompt, instantly generating a full leaderboard over all models without requiring model responses to be generated. This yields efficient leaderboard creation over arbitrary prompt sets.

**Meta-learning.** P2L is related to meta learning (Schmidhuber, 1987; Santoro et al., 2016; Finn et al., 2017) insofar as we are training a model to output models. For example, we have discussed training an LLM (the meta-learner) to output coefficients of a BT regression (the learner). However, the meta-learning literature primarily focuses on learners that are deep neural networks. Instead, we let the learner be an extremely simple statistical model that is used for inference.

**Routing.** Prior work on routing LLM queries optimizes trade-offs between cost and performance, typically through classifiers or gating mechanisms. RouteLLM (Ong et al., 2024) and AutoMix (Madaan et al., 2023), Hybrid LLM (Ding et al., 2024), train classifiers to decide between a strong and weak model, while LLM-Blender (Jiang et al., 2023) ranks candidate responses and blends them. RouterDC (Chen et al., 2024) uses contrastive losses to train a query-based router. Unlike these approaches, which operate over a small fixed set of models, P2L learns a parametric function mapping prompts to full model leaderboards. Its statistical structure supports efficient cost-aware routing, outperforming static models in live crowdsourced settings while scaling to personalized and task-specific selections.

**Benchmark Compression.** P2L can be related to benchmark compression (Polo et al., 2024). Rather than reducing benchmark size via pruning data examples, P2L captures and compresses benchmark information *parametrically*.

**Parametric statistical models.** Our work builds on classic log-linear models and GLMs, like those of Bradley and Terry (1952); Rao and Kupper (1967); see (McCullagh, 2019) for a review, and (Ameli et al., 2024) for further extensions that enrich this model class for better LLM ranking. The closest piece of work to ours is Hastie and Tibshirani (1993), which proposes varying-coefficient models. P2L can be seen as a subclass of varying-coefficient models. To our knowledge, ours is the first work to parameterize such a model via a foundation model and backpropagate it end-to-end, while the techniques in Hastie and Tibshirani (1993) use bespoke fitting procedures and simpler statistical models than LLMs.

## Impact Statement

This paper presents work whose goal is to advance the field of Large Language Model evaluation. There are many potential societal consequences of our work, none which we feel must be specifically highlighted here.

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

# A. Proofs

*Proof of Theorem 1.* The equivalence of (4) and (6) is immediate. Proving the equivalence of (5) and (6) is more challenging, and we focus there.

We begin by simplifying the expressions in (5). The cost constraint can be succinctly written as $\tilde{\pi}^\top c \leq C$. Regarding the objective, because the binary cross-entropy loss is linear in the response,

$$
\mathbb{E}_{B \sim \tilde{\pi}, A \sim q, Y' \sim \text{Bern}(\sigma(\theta^*(z)_B - \theta^*(z)_A))} \left[ \ell(\sigma(\theta - \theta^*(z)_A), Y') \mid Z = z \right]
$$
$$
= \mathbb{E}_{B \sim \tilde{\pi}, A \sim q} \left[ \ell(\sigma(\theta - \theta^*(z)_A), \sigma(\theta^*(z)_B - \theta^*(z)_A)) \mid Z = z \right]
$$
$$
= \mathbb{E}_{A \sim q} \left[ \ell \left( \sigma(\theta - \theta^*(z)_A), \left(\tilde{\pi}^\top \mathbf{W}^*\right)_A \right) \,\middle|\, Z = z \right],
$$

where again $\mathbf{W}^*$ represents the population win matrix, with entries $\mathbf{W}^*_{ba} = \sigma(\theta^*(z)_b - \theta^*(z)_a)$. Thus, the optimization problem in (5) can be equivalently rewritten as

$$
\underset{\tilde{\pi} \in \Delta^M}{\text{maximize}} \quad \theta'(\tilde{\pi}) \quad \text{subject to} \quad \tilde{\pi}^\top c \leq C,
$$

where

$$
\theta'(\tilde{\pi}) = \underset{\theta \in \mathbb{R}}{\text{argmin}} \; \mathbb{E}_{A \sim q} \left[ \ell \left( \sigma(\theta - \theta^*(z)_A), (\tilde{\pi}^\top \mathbf{W}^*)_A \right) \right].
$$

Examining the first-order conditions of the inner optimization problem for $\theta'(\tilde{\pi})$ shows that the solution satisfies

$$
\sum_A q_A \, \sigma\left(\theta'(\tilde{\pi}) - \theta^*(z)_A\right) = \tilde{\pi}^\top \mathbf{W}^* q. \tag{9}
$$

Define

$$
R(\tilde{\pi}) = \tilde{\pi}^\top \mathbf{W}^* q, \qquad G(\theta) = \sum_A q_A \, \sigma(\theta - \theta^*(z)_A).
$$

Then $\theta'(\tilde{\pi}) = G^{-1}(R(\tilde{\pi}))$. Since $G^{-1}$ is strictly increasing,

$$
\underset{\tilde{\pi}}{\text{maximize}} \, \theta'(\tilde{\pi}) \quad \Longleftrightarrow \quad \underset{\tilde{\pi}}{\text{maximize}} \, R(\tilde{\pi}).
$$

Thus, the problem reduces to:

$$
\underset{\tilde{\pi} \in \Delta^M, \, \tilde{\pi}^\top c \leq C}{\text{maximize}} \, \tilde{\pi}^\top \mathbf{W}^* q,
$$

which is exactly the problem in (6). $\qquad\square$

# B. Additional theory

## B.1. Aggregating leaderboards via averaging

The BT model tells us that for all $z \in \mathbb{Z}$,

$$
\log \left( \frac{\mathbb{P}(Y = 1 \mid X = x, Z = z)}{1 - \mathbb{P}(Y = 1 \mid X = x, Z = z)} \right) = x^\top \theta^*(z).
$$

Thus,

$$
\mathbb{E}_{Z \sim Q} \left[ \log \left( \frac{\mathbb{P}(Y = 1 \mid X = x, Z)}{1 - \mathbb{P}(Y = 1 \mid X = x, Z)} \right) \right] = x^\top \left( \underbrace{\int_{z \in \mathcal{Z}} \theta^*(z) dQ(z)}_{\tilde{\theta}(Q)} \right).
$$

That is, taking a (weighted) average of the values of $\theta^*(z)$ leads to a predictor of the expected log-odds.

This method has two downsides: firstly, increasing the $m$th coordinate of $\tilde{\theta}(Q)$ does not mean that model $m$ is more likely to win against other models on average. Secondly, the function $\tilde{\theta}(Q)$ does not have a simple relationship with the win rate. This motivates the need for the aggregation metric from Section 2.1.1.

## C. Additional Routing Figures

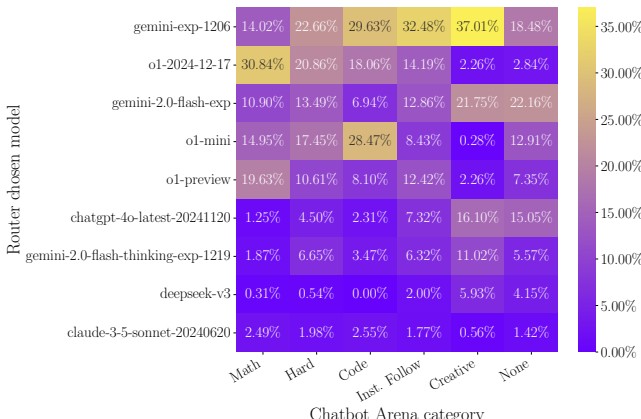

Figure 7: **Router model choice distribution** in each prompt category. The rows are different models, and the columns are different categories. Each cell represents the probability that the model was selected within that category (i.e., columns sum to 1). Models with an average selection rate below 1% are not shown.

## D. Additional regression tests

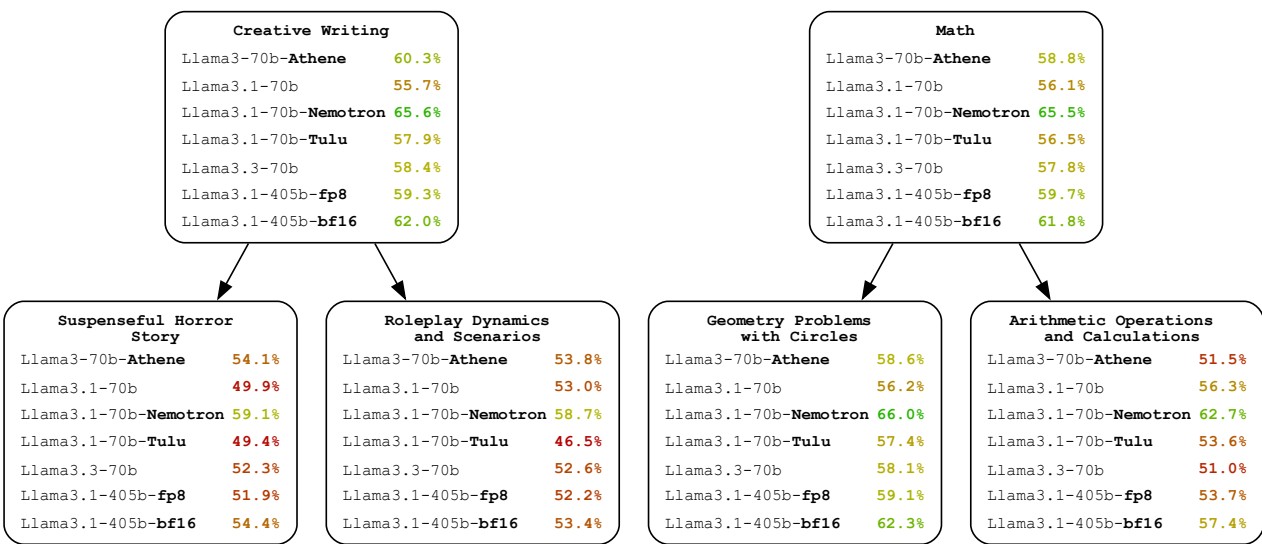

Figure 8: **Regression test** on Llama models with creative writing and math prompts. The percentages shown signify win rates against `Llama-3-70B` under the BT coefficients predicted from `P2L-7B`.

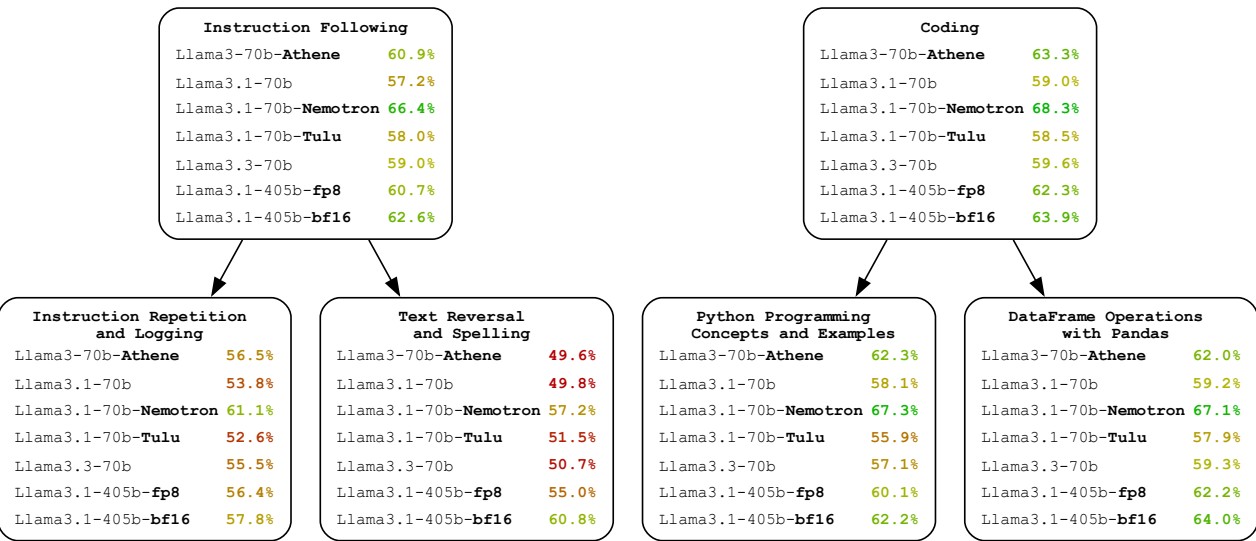

Figure 9: **Regression test** on Llama models with instruction following and coding prompts. The percentages shown signify win rates against `Llama-3-70B` under the BT coefficients predicted from `P2L-7B`.

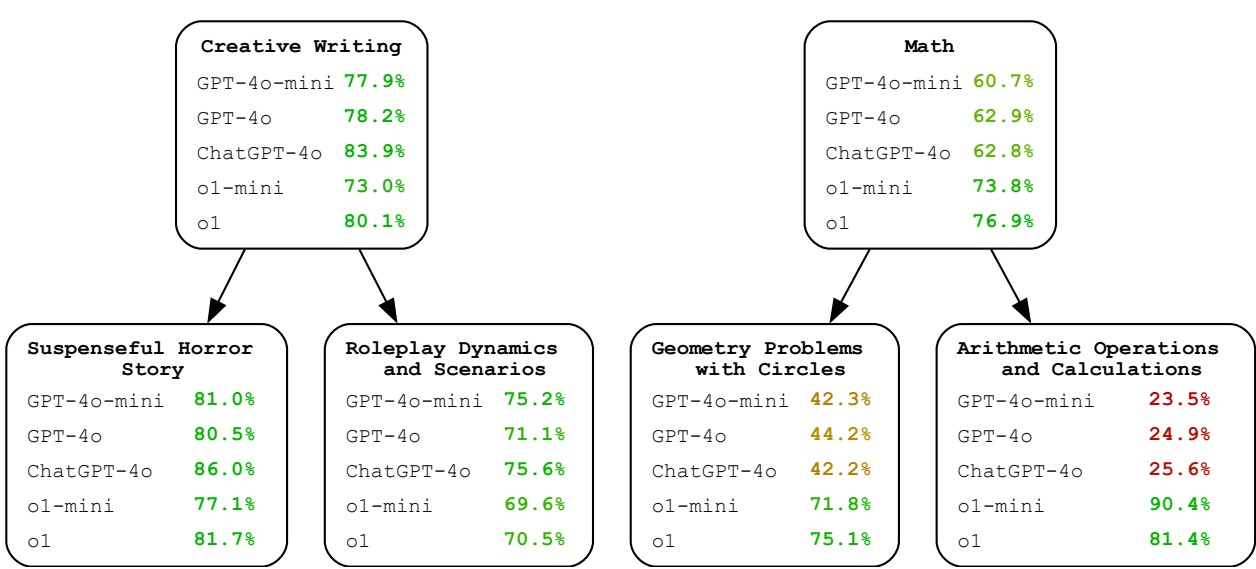

Figure 10: **Regression test using grounded Rao-Kupper.** We show the strengths of different OpenAI models on various topic clusters based on `P2L-7B` with a grounded RK regression head (see Section 2.2) and a dataset of unlabeled prompts. The percentage represents the sigmoid of the model coefficient. Because the RK model is grounded, this corresponds roughly to a signal of the model's reliability, i.e., its tendency to produce an answer that exceeds the voter's minimum bar of quality. The results show strong category-specific variability in performance; for example, `GPT-4o-mini` and `o1` have roughly the same reliability in the category "Suspenseful Horror Story", but not "Arithmetic Operations and Calculations". We can also see that some categories are more difficult in general for LLMs to answer reliably, and thus we see larger performance improvements from test-time compute models like `o1` and `o1-mini`.

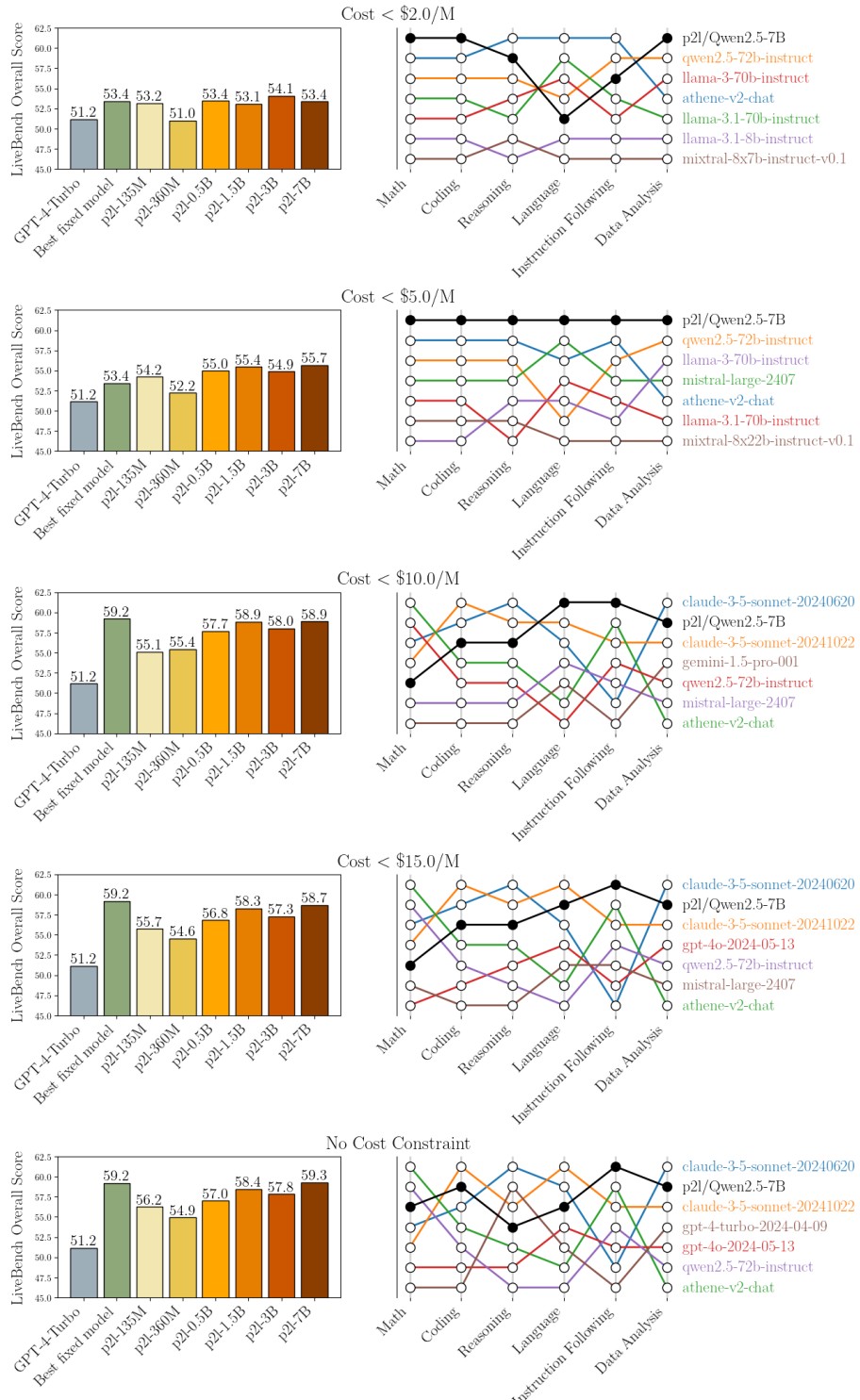

Figure 11: **LiveBench cost routing.** Comparison of the P2L cost-aware router and static models on LiveBench under various inference-cost constraints. The left plots show each model's overall LiveBench performance at different maximum cost thresholds, while the right plots display models' relative rankings across multiple categories at the specific cost limit. By adaptively allocating prompts to cheaper or more expensive models when advantageous, the P2L router consistently matches or surpasses the best single model within each budget.

| Model | LiveBench Score | Math | Coding | Reasoning | Language | Instruction Following | Data Analysis |
|---|---|---|---|---|---|---|---|
| P2L-7B | **59.3** | 51.9 | 65.2 | 50.0 | 56.5 | **75.8** | 56.3 |
| claude-3-5-sonnet-20240620 | 59.2 | 51.3 | 63.5 | **54.7** | 56.8 | 72.3 | **56.7** |
| claude-3-5-sonnet-20241022 | 59.0 | 51.3 | 66.8 | 50.0 | **57.0** | 74.1 | 54.9 |
| P2L-1.5B | 58.4 | **55.3** | **67.5** | 48.0 | 51.4 | 71.9 | **56.7** |
| P2L-3B | 57.8 | 49.6 | 66.8 | 50.7 | 53.3 | 70.4 | 56.2 |
| P2L-0.5B | 57.0 | 51.9 | 59.6 | 50.7 | 51.7 | 73.4 | 54.8 |
| P2L-135M | 56.2 | 48.9 | 63.5 | 50.0 | 47.1 | 74.1 | 54.0 |
| P2L-360M | 54.9 | 52.4 | 58.1 | 44.0 | 44.1 | 74.4 | **56.7** |
| athene-v2-chat | 53.4 | 53.4 | 56.9 | 48.0 | 37.5 | 74.6 | 50.2 |
| gpt-4o-2024-05-13 | 52.8 | 42.7 | 50.4 | 47.3 | 49.3 | 72.4 | 54.4 |
| qwen2.5-72b-instruct | 52.6 | 52.3 | 55.6 | 47.3 | 36.0 | 73.3 | 51.1 |
| gpt-4-turbo-2024-04-09 | 51.2 | 40.3 | 45.8 | 52.7 | 45.3 | 68.4 | 54.5 |
| mistral-large-2407 | 50.4 | 48.4 | 45.8 | 44.0 | 40.5 | 73.1 | 50.4 |
| chatgpt-4o-latest-20241120 | 49.4 | 37.7 | 44.4 | 44.7 | 43.7 | 74.1 | 51.7 |
| gemini-1.5-pro-001 | 44.2 | 36.2 | 33.7 | 34.0 | 37.6 | 68.9 | 54.8 |
| llama-3.1-70b-instruct | 42.4 | 34.4 | 32.9 | 34.7 | 36.4 | 68.9 | 47.3 |
| llama-3-70b-instruct | 41.7 | 26.3 | 28.7 | 40.0 | 36.3 | 68.5 | 50.7 |
| mixtral-8x22b-instruct-v0.1 | 37.5 | 28.0 | 32.3 | 36.0 | 27.9 | 65.5 | 35.5 |
| llama-3.1-8b-instruct | 26.3 | 19.5 | 14.5 | 18.7 | 17.8 | 53.9 | 33.3 |
| mixtral-8x7b-instruct-v0.1 | 22.1 | 12.4 | 10.6 | 23.3 | 12.8 | 46.1 | 27.4 |

Table 1: **LiveBench performance comparison**. Comprehensive evaluation of language models across seven capability categories: overall LiveBench score, mathematics, coding, reasoning, language understanding, instruction following, and data analysis. Results show performance comparison between p2l models at different parameter scales (135M to 7B), Claude-3.5 Sonnet versions, and other leading language models including GPT-4, Gemini, and LLaMA variants. All models were evaluated using identical inference settings as those employed in Chatbot Arena to ensure fair comparison. Scores are presented as percentages, with the highest score in each category shown in **bold** and second-highest underlined. P2L-7B achieves top performance in LiveBench Score (59.3) and Instruction Following (75.8), while maintaining competitive performance across other categories.

# E. Additional information

## E.1. Clustering Method

We leverage a topic modeling approach using BERTopic. We first encode each prompt using OpenAI's embedding model, `text-embedding-3-small`, reduce dimensions with UMAP, and apply a hierarchical-based clustering algorithm (HDBSCAN) with min size cluster 8. This process generates distinct topic clusters. Each topic is then summarized and named using an LLM (GPT-4o-mini). This process is replicated from ArenaHard's clustering pipeline (Li et al., 2024).

## E.2. Accuracy

| Model | Accuracy (%) |
|---|---|
| Random | 25.00 |
| Marginal | 37.40 |
| 0.135B | 40.42 |
| 0.36B | 42.23 |
| 0.5B | 46.06 |
| 1.5B | 47.06 |
| 3B | 47.41 |
| 7B | 47.88 |

Table 2: Grounded Rao-Kupper model accuracy by parameter size. Accuracy is calculated according to correctly predicting the classes: `{win, loss, tie, tie (both bad)}`.

## E.3. Training Costs

P2L models are fairly inexpensive to train: P2L-7B on 1.5 million data points costs less than $250 to train end-to-end using a relatively unoptimized Deepspeed and Huggingface Trainer infrastructure ($23.92 per hour for 8xH100 on Runpod). The well-performing 3B and 1.5B variants train with negligible cost.

## E.4. Model list

The full list of models is: `athene-v2-chat` (Frick et al., 2024), `chatgpt-4o-latest-20241120`, `claude-3-5-haiku-20241022`, `claude-3-5-sonnet-20240620`, `claude-3-5-sonnet-20241022` (Anthropic, 2024), `deepseek-v3` (Liu et al., 2024), `gemini-1.5-flash-001`, `gemini-1.5-flash-002`, `gemini-1.5-pro-001`, `gemini-1.5-pro-002` (Team et al., 2024a), `gemini-2.0-flash-exp`, `gemini-2.0-flash-thinking-exp-1219`, `gemini-exp-1206`, `gemma-2-27b-it`, `gemma-2-9b-it` (Team et al., 2024b), `glm-4-plus`, `gpt-4-1106-preview`, `gpt-4-turbo-2024-04-09` (OpenAI, 2023), `gpt-4o-2024-05-13`, `gpt-4o-2024-08-06`, `gpt-4o-mini-2024-07-18` (OpenAI, 2024), `llama-3-70b-instruct`, `llama-3.1-405b-instruct-fp8`, `llama-3.1-70b-instruct`, `llama-3.1-8b-instruct`, `llama-3.3-70b-instruct` (AI@Meta, 2024), `mistral-large-2407`, `mixtral-8x22b-instruct-v0.1`, `mixtral-8x7b-instruct-v0.1` (Jiang et al., 2024), `o1-2024-12-17`, `o1-mini`, `o1-preview` (Jaech et al., 2024), `qwen2.5-72b-instruct` (Team, 2024), and `yi-lightning` (Young et al., 2024).

