# OpenReview forum: "Prompt-to-Leaderboard: Prompt-Adaptive LLM Evaluations"
_ICML.cc/2025/Conference — ICML 2025 poster_

### Official Review · Reviewer_Xq7j · 2025-02-21

**Overall Recommendation:** 4

**Summary:**

Language model evaluations typically rely on aggregated metrics like accuracy or human preference, averaging across users and prompts. This averaging obscures user- and prompt-specific variations in model performance. The authors propose Prompt-to-Leaderboard (P2L), a method that predicts prompt-specific leaderboards by training large language models on ChatBot Arena preference data and outputting predicted scores for each model in ChatBot Arena, for each input prompt. The authors train routers based on this methodology and achieve the #1 spot in the Chatbot Arena leaderboard.

**Claims And Evidence:**

I think this paper's contributions are sound and would be of general interest, but I have some concerns about the clarity of the presentation and the rigor of the conclusions which I would like to see addressed. The authors overclaim in several places, which is unnecessary, as the work is sufficiently interesting without hyperbole.

In Sec. 3.6, I appreciate the authors' inclusion of a second, so-called "out of distribution" benchmark to evaluate their method; LiveBench is a reasonable choice here. However, the analysis of the authors' results considerably oversells the reality. The obvious naive baseline for their comparatively computationally intensive, complex and expensive method is to simply use the best performing model on the leaderboard for every prompt. In the case of LiveBench, this trivial baseline is either better than, or statistically no different from, P2L, when not controlling for cost. This is a very important limitation, and I would like to see the authors acknowledge it as such, and address why this may be the case. Their point about the cost-effectiveness of P2L in this setting is more compelling, and should be retained.

**Essential References Not Discussed:**

The authors should document in their related works the considerable research on benchmark compression which has emerged lately. It is at this point well understood that even for carefully curated benchmarks, only a small subset of the entire benchmark is necessary to establish stable model rankings (https://arxiv.org/abs/2402.14992). Aggregated compressed benchmarks such as (https://mixeval.github.io/) are relatively inexpensive to curate and run, compared to P2L, and correlate well with LMSYS Chatbot Arena.

**Experimental Designs Or Analyses:**

The categorization mechanism described in 3.4 is superfluous; LiveBench already includes these categorizations, ChatBot Arena allows reranking according to a wide range of types (https://lmarena.ai/), and many prior works anticipate the strategy, so it is not novel. This could be relegated to the appendix.

It's not clear to me how the claim "P2L’s predictions over singular prompts differ more drastically from category leaderboards" is supported by Fig. 7.

The authors make frequent use of a particular type of figure, which reports distinct model rankings for topic clusters. These figures take up a lot of space in the paper, and I find the results themselves puzzling. In Fig. 6, o1 is the best for all math-related tasks, and ChatGPT-4o is the best model for all other kinds of tasks. This distinction was discussed by OpenAI and many others, and it doesn't require an LLM to guess that this would be the outcome. In Fig. 8 and Fig. 9, Nemotron is the best on every subset, an even less interesting result. In short, I don't understand how these summary figures are useful; I would have much preferred to have had a link to raw model outputs for each benchmark, so that I could have evaluated the results myself.

**Methods And Evaluation Criteria:**

In Related Work, the authors claim that P2L is unlike prior routing approaches; is this intended to justify the fact that no baseline methods are included in several key experimental figures, such as 3 and 7, and the baselines in 2 are weak, by the authors' own acknowledgement? A few more cheap-to-evaluate baselines, like decision trees over embeddings of the prompts, or a bag-of-words approach, would increase my confidence in the authors' claims that their method is valuable.

**Other Comments Or Suggestions:**

In general, the figure captions in the paper are inadequate. Figure captions should be expanded to give all the details necessary to understand the figure, and where those details are too extensive, should include hyperlinks to the relevant sections of the paper.

**Other Strengths And Weaknesses:**

I have no other strengths or weaknesses to note.

**Questions For Authors:**

None.

**Relation To Broader Scientific Literature:**

The idea of a reward model going from prompts directly to leaderboards is, as far as I know, novel.

**Theoretical Claims:**

I reviewed the theoretical claims in the main paper and I have no concerns.

---

> ### Author Rebuttal · Authors · 2025-04-01
>
> We sincerely thank the reviewer for the thoughtful and constructive feedback. We address your concerns and propose corresponding revisions:
>
>
> **W1 OOD Results:** We understand that using the top static model might appear an intuitive baseline. However, practically, this static model is unknown ahead of time, and P2L’s value is precisely in selecting the best-performing model dynamically.
> This means P2L, which never sees ground truth labels or model responses, performs as good as running all models on LiveBench, scoring them using the benchmark’s ground truth labels, and selecting the best model *after the fact*.
> We will communicate this more clearly in future revisions.
>
> **W2 Baselines:** During development, we attempted non-deep learning approaches such as KNN and embedding based methods. We found these did not perform better than, and sometimes worse than, the marginal baseline– with a log-loss delta of less than 0.01. Ultimately, we chose the marginal BT as baseline, since this was the most effective and stable baseline– we will ensure to communicate this in the revision. In the end, the goal of this work is to create a scalable (along data and parameter count) method to provide granular model evaluations, which our deep learning approach provides.
>
> **W3 Clustering Novelty:** We are not claiming any novelty in the clustering algorithm. We are claiming that ours is the first approach that allows us to create a leaderboard for small, possibly singular, clusters, moreover doing so label-free. We will clarify in the final revision that our clustering mechanism is for demonstrative purposes and will relegate extensive discussions to supplementary materials.
>
> **W4 Fig 7:** We recognize our current presentation was insufficiently clear. We will enhance the caption of Fig. 7 to explicitly state what the function distance represents, how it relates to prompt-specific leaderboards, and how this concretely supports our claim. Specifically, the large function distance between P2L’s leaderboard and the marginal leaderboards for a smaller number of prompts (<= 10) suggest P2L's predictions differ from the marginal.
>
> **W5 Uninteresting Results:** We appreciate the reviewer’s feedback on Figures 6, 8, and 9, and agree that some of these results align with existing expectations—such as o1 performing well on mathematical tasks and ChatGPT-4o excelling in creative tasks. However, we argue that these confirmatory results are precisely what demonstrate the reliability and effectiveness of P2L. Moreover, P2L systematically captures these known performance trends without requiring expensive per-prompt annotations, thus validating our approach as robust and practically useful for model selection and routing. Additionally, our finding that GPT-4o-mini matches or surpasses substantially more expensive (up to 100x) models (e.g., o1 or GPT-4o) in certain prompt clusters highlights an important, practically valuable insight into cost-effective model utilization that is only clearly revealed by our method. We will revise the manuscript to more explicitly emphasize these practical insights, clarify how they support the validity of our approach, and ensure the results' practical implications are clearly communicated.
>
> **W6 Public Release:** We will release all P2L models, routing code, training code, evaluation code publicly. We will host a P2L endpoint to provide a way for readers to calculate raw P2L outputs for any prompt of interest.
>
> **W7 Related Work:** We will elaborate our connection to benchmark compression in related work. P2L uses deep learning to compress evaluation signals parametrically, uniquely allowing it to be label-free at test time.
>
> **W8 Figure Clarity:** We will comprehensively revise all figure captions in the revision, ensuring they succinctly summarize methods, key findings, and implications. Where necessary, we will include hyperlinks to relevant text.
>
> ---
>
> We greatly appreciate the reviewer’s constructive feedback, which significantly enhances the quality and clarity of our work. We respectfully ask the reviewer to reconsider their rating, given our revisions, clarifications, and the substantial potential impact of our contributions to the community.

---

### Official Review · Reviewer_EY29 · 2025-03-12

**Overall Recommendation:** 3

**Summary:**

This paper introduces Prompt-to-Leaderboard (P2L), a method for generating prompt-specific leaderboards of large language models (LLMs) by training LLMs on human preference data. The core idea is to model prompt-dependent Bradley-Terry coefficients, enabling per-prompt performance comparisons. Key applications include optimal query routing, personalized evaluations, and automated model strength/weakness analysis. The authors validate P2L on Chatbot Arena and LiveBench, demonstrating that P2L routers outperform static models in live evaluations and generalize well to out-of-distribution tasks. Scaling experiments suggest P2L follows power-law improvements with model size and data.

**Claims And Evidence:**

The assertion that P2L "achieves #1 on Chatbot Arena" (Section 3.3.1) lacks transparency: How was the Arena score computed?

**Essential References Not Discussed:**

NA

**Experimental Designs Or Analyses:**

The power-law trend (Section 3.2) is convincing.

However, simulated costs (Section 3.3.2) rely on token-length averages, ignoring variance in real deployments.

**Methods And Evaluation Criteria:**

Methods: The use of parametric regression coefficients (e.g., BT) is appropriate for modeling pairwise preferences. Extensions to ties and "both bad" scenarios are innovative.

Evaluation: LiveBench and Chatbot Arena are standard benchmarks. However, category-specific leaderboards (Section 3.4) rely on automated clustering without human validation. It may result in noisy or subjective categories, and lead to ambiguous categorization criteria.

**Other Comments Or Suggestions:**

NA

**Other Strengths And Weaknesses:**

- The paper combines parametric statistics with LLMs for prompt-adaptive evaluation, which is novel and impactful.

- The paper is well-structured but dense; more visualizations would improve accessibility.

**Questions For Authors:**

What explains the outlier performance of P2L-1.5B in Figure 3?

**Relation To Broader Scientific Literature:**

- Builds on RLHF and BT models but innovates by integrating prompt conditioning.

- Contrasts with routing methods (e.g., RouteLLM, AutoMix) by enabling large-scale, cost-aware routing.

**Theoretical Claims:**

Theorem 1’s proof (Appendix A) assumes ideal BT conditions and ignores non-transitivity. Practical deviations (e.g., model ties) could invalidate equivalence.

---

> ### Author Rebuttal · Authors · 2025-04-01
>
> We sincerely thank the reviewer for the thoughtful and constructive feedback. We address your concerns and propose corresponding revisions:
>
> **W1 Arena Score Computation:** We deploy the P2L router onto Chatbot Arena, routing between the models detailed in Appendix D1. We collect blind human preference votes between responses from the P2L router against active public models on Chatbot Arena. We use these preference votes to calculate the Bradley-Terry regression over all Chatbot Arena battles to produce an Arena Score. This is the standard method to add a model to the Chatbot Arena leaderboard. We will ensure Arena Score calculations are well documented in future revisions. [This Google Colab Notebook](https://colab.research.google.com/drive/1KdwokPjirkTmpO_P1WByFNFiqxWQquwH) details exactly how Arena Scores are computed on the Chatbot Arena leaderboard. Our results, including comparison vote data, will be publicly available after double blind is lifted.
>
> **W2 Categorization:** We acknowledge that the automated clustering strategy has risk for noise. Note P2L can create an aggregate leaderboard over any subset of input prompts; this is detailed in section 2.1.1. Thus, any clustering strategy can be employed with P2L– our paper only details one such strategy as an example. We believe that co-developing clustering methods with P2L based rankings is an interesting future research direction.
>
> **W3 Theoretical Assumptions:** The reviewer is correct, and we try to be clear about this in the paper. The proof says that under the Bradley-Terry model, these routers are equivalent. We explicitly say, after the theorem statement, that “It is important to note that deviations from the BT model—for example, any non-transitivity—will break this relationship" (line 162).
> Thus, we believe this limitation has already been sufficiently communicated.
>
> **W4 Cost Estimation:** This is correct. We will mention this limitation explicitly in the paper.
>
> **W5 Paper Density:** We thank the reviewer for feedback. We will aim to increase accessibility in future revisions with intuitive visualization illustrating P2L’s functionality.
>
> **Q1:**
> > What explains the outlier performance of P2L-1.5B in Figure 3?
>
> We attribute this to noise. The bootstrapped 95% confidence intervals overlap, indicating this variation is well-within sampling variability. We view the trend in the plots as more informative than the specific values.
>
> ---
>
> We greatly appreciate the reviewer’s constructive feedback, which significantly enhances the quality and clarity of our work. We respectfully ask the reviewer to reconsider their rating, given our revisions, clarifications, and the substantial potential impact of our contributions to the community.

---

### Official Review · Reviewer_d5DE · 2025-03-21

**Overall Recommendation:** 3

**Summary:**

This paper proposes a method that routes a prompt to a specific LLM from a given LLM list. Given a dataset of various prompts, responses from different models and the pairwise preference result, the method train a mapping from prompt to feature that fit the reward gain by the model when fed with the prompt. Then when given a new prompt, the method can be used to predict the most suitable LLM for this prompt. Experiment shows that the proposed method outperforms the best single model on Chatbot Arena.

**Claims And Evidence:**

The P2L model, when doing optimal routing, outperform single models. This claim is properly consolidated by the experiments in presented in the paper.

**Essential References Not Discussed:**

No significant missing of references

**Experimental Designs Or Analyses:**

This paper seems lack some details in experiment setup, which make paper hard to follow

1. What is the training parameters when training the P2L models?

2. How is the evaluation on Chatbot Arena conducted? Also the author claims that they deployed P2L on Chatbot Arena but I didn't find a model called "P2L" on the [leaderboard](https://lmarena.ai/?leaderboard). Could the authors provide more detailed information?

3. In section 3.4, how did the authors map a prompt to categories? Also, to conduct hierarchical clustering, how did the authors define the distance between different categories?

**Methods And Evaluation Criteria:**

My concerns regarding the method are listed as follows

1. It looks like that the method is not scalable enoough. For example, initially we have 30 models. According to Section 3.1, we are initializing the coefficient head to map to $\mathbb{R}^{30}$. However, if now we need to add one more model to our model list, then we need the coefficient head to be a mapping with co-domain $\mathbb{R}^{31}$, meaning that we need to re-train all the parameters from scratch. This issue is especially severe given P2L requires 1.5M training data.

2. Given a prompt, the P2L framework requires first using the P2L model (backboned by an LLM) to compute the leaderboard. This introduces additional computational overhead and might be time consuming. It is doubtable whether this sacrifice is worthy the performance gain from selecting a (potentially) more suitable LLM given that the overall improvement brought by P2L is not significant.

Also, the paper employs two evaluation criteria for optimal routing. Some of my concerns are listed below

1. First, the paper study the generalization of feedback prediction on a hold-out validation set from Chat-bot Arena and report the validation loss and squared error. This is not straightfoward enough because the calculation of the loss is not clearly stated in the paper. A more straightforward way is to report the validation accuracy, which is not reported in the paper

2. Secondly, the paper considered Chatbot Arena as the benchmark for the optimal routing. While Chatbot Arena is a widely-recognized benchmark for alignment, it is unclear how the author deploy and test their model with Chatbot Arena

**Other Comments Or Suggestions:**

N/A

**Other Strengths And Weaknesses:**

N/A

**Questions For Authors:**

N/A

**Relation To Broader Scientific Literature:**

This paper is related to the broad related works focusing on LLM routing, to which I am not very familiar.

**Theoretical Claims:**

The proof of the theorems looks correct although I didn't check very detailedly.

---

> ### Author Rebuttal · Authors · 2025-04-01
>
> We sincerely thank the reviewer for the thoughtful and constructive feedback. We address your concerns and propose corresponding revisions:
>
> **W1 Scalability**: We agree with the reviewer that reducing the cost of adding new models is of interest, and are excited to continue exploring methods, such as online learning, to optimize this in future work. However, we also note our P2L models are fairly inexpensive to train: P2L-7B on 1.5 million data points costs less than \\$250 to train end-to-end using a relatively unoptimized Deepspeed + Huggingface Trainer infrastructure (\\$23.92 per hour for 8xH100 on Runpod). The well-performing 3B and 1.5B variants train with negligible cost. We will include these exact training hardware, time, and cost numbers in the appendix for future revisions, which we believe will clarify cost concerns.
>
> **W2 Cost**: The cost of the P2L model is negligible, both in terms of compute and time. P2L is, at its largest, a 7B model inferencing a single forward pass on the prompt only, which is very fast: P2L 7B add around 5% overhead on first token latency on an A40, $0.40 per hour on Rundpod, and less than 1% of the average routed LLM’s cost. The performance gain, on the other hand, is substantial. Therefore we do not see this point as a concern in practice. We will update the paper to communicate this fact.
> W3 Validation Metrics: We will include clear full definitions detailing the loss calculation for additional clarity in future revisions. Moreover, we will include straight forward accuracy metrics in the next revision. For context, we have computed accuracies for the grounded RK models which classify {A, B, Tie, Tie Both Bad}:
>
>
> | Model     | Accuracy (%) |
> |-----------|--------------|
> | Random    | 25.00        |
> | Marginal  | 37.40        |
> | 0.135B    | 40.42        |
> | 0.36B     | 42.23        |
> | 0.5B      | 46.06        |
> | 1.5B      | 47.06        |
> | 3B        | 47.41        |
> | 7B        | 47.88        |
>
> **W4 Deployment Clarity**: We will update our revised paper to include greater detail on deployment to Chatbot Arena. Specifically, we currently detail the P2L router’s model list in Appendix D1. We will additionally specify that we collect blind pairwise comparisons against all active public models hosted on Chatbot Arena in a process identical to how standard models are added to the Chatbot Arena Leaderboard.
>
> **W5 Training Params**: When training P2L we do full parameter training. This means we update the weights of both the pretrained transformer and the newly initialized linear head. We will ensure this section is fully specified in section 3.1 in future revisions.
>
> **W6 Missing from Leaderboard**: It was deployed in battle mode on Chatbot Arena for some time, and with that data, we are able to calculate its leaderboard position even though it is not displayed. The model does not appear on the Chatbot Arena leaderboard because routers are not allowed on the leaderboard– however, results, including comparison vote data, will be publicly available after double blind is lifted.
>
> **W7 Clustering Method**: We leverage a topic modeling approach using BERTopic. We first encode each prompt using OpenAI’s embedding model, text-embedding-3-small reduce dimensions with UMAP, and apply a hierarchical-based clustering algorithm (HDBSCAN) with min size cluster 8. This process generates distinct topic clusters. Each topic is then summarized and named using an LLM (GPT-4o-mini).
>
> ---
>
> We greatly appreciate the reviewer’s constructive feedback, which significantly enhances the quality and clarity of our work. We respectfully ask the reviewer to reconsider their rating, given our revisions, clarifications, and the substantial potential impact of our contributions to the community.

---

### Official Review · Reviewer_Hdbw · 2025-03-25

**Overall Recommendation:** 2

**Summary:**

This paper is motivated by the fact that LLMs are sensitive to prompts, and current benchmarks such as Chatbot Arena leverage pair-wise comparisons from users to rank models without explicitly controlling the prompt distribution. The paper proposes a method that essentially trains a model to predict the "model advantages" vector from a collection of tuples (prompt, two-hot vectors). The authors further consider ties in human preferences to gain better and more precise signals for interpreting model abilities with different prompts. The trained models vary in different sizes and have shown good performance on the leaderboard as well as a balance between performance and cost.

**Claims And Evidence:**

Evidence is well-supported. However, the motivation in the introduction is rather confusing at first. After "In other words, P2L models take a prompt as input and output a leaderboard of LLMs for that specific prompt," the authors don't explain further about the meaning of "leaderboard of LLMs" but rather continue by stating "The P2L model can be trained based on any feedback signal, for example, binary human preference data..." I was able to understand the meaning of "prompt to leaderboard" when I got to the method section, but it's not clear before that.

**Essential References Not Discussed:**

This paper comes after the ICML submission deadline but is highly relevant: https://arxiv.org/pdf/2502.14815

While experimemtal comparison is hard and there's difference in terms of the settings for sure, I am wondering the advantage of P2L over https://arxiv.org/pdf/2502.14815 in the methodology level.

Another line of relevant work may be mixture-of-expert models and particularily similar training paradigms as the method introduced in the paper.

**Experimental Designs Or Analyses:**

The experiments include quantitative results from P2L router performance and cost study.

**Methods And Evaluation Criteria:**

The evaluation is done with a comparison between P2L and multiple prevailing models, especially the closed-source ones.

**Other Comments Or Suggestions:**

line 80: from Z->R^M => \theta: Z->R^M

**Other Strengths And Weaknesses:**

The method is clear, but the writing is a bit hard to follow with ~10 variables defined and mixed with the main text. Some terminologies could be made more official and easier for people to follow.
The experiments seem to be adequate and come along with a nice clustering study in Figure 6, and the cost-performance trade-off study in Figure 5.

The problem setting itself is novel to me. However, I am not so sure about novelty at the method level, and it would help if the author could provide some clear explanations and comparison with existing works, not limited to the field of model routing.

Moreover, the main thing that is not clear for me is that some more intuitive baselines should be compared with to justify the model training. For example, given a dataset {(p_i, z_i) | i=1,...}, where p_i is the prompt and z_i is the two-hot vector from human preference. Given a new user prompt p, why not retrieve the top k (e.g., k=100) similar prompts, say {p1, ..., p100} as well as their zs {z1, ..., z100} from the dataset (let the similarity between p and p_i be s_i) and use the weighted score \sum z_i * s_i as the predicted final performance. How does it compare with P2L?

These being said, the problem itself is interesting and the experiments seem to be solid. I will consider adjusting my scores if the concerns above are addressed in some way.

**Questions For Authors:**

See above.

**Relation To Broader Scientific Literature:**

The key idea of modeling the probabilistic distribution based on model preference seems to align with the key idea in mixture-of-expert models, i.e., the training of the gating model. Particularly, one can have dense aggregation or sparse aggregation (max pool) given the gating model's output. The main difference seems to be in the problem setting where human preference, i.e., two-hot encoding, is given, which requires the fitting schema introduced in the core method.

**Theoretical Claims:**

I've read through the theories in the main paper, I believe they are correct based on my knowledge.

---

> ### Author Rebuttal · Authors · 2025-04-01
>
> We sincerely thank the reviewer for the thoughtful and constructive feedback. We address your concerns and propose corresponding revisions:
>
> **W1 Intro**: We will revise our introduction to explicitly define 'leaderboard of LLMs' as a prompt-specific ranking of multiple LLMs to improve clarity.
>
> **W2 LLMSelector**: We believe LLMSelector is an excellent work which shares some philosophies with P2L. Importantly, it is concurrent work, and we should not be responsible for establishing novelty over LLMSelector.
> Additionally,  the goals of the two works are different. LLMSelector aims to select models in compound AI systems using an LLM judge. Instead, P2L’s methodology instead gives full rankings of chat models for a single input.
>
> **W3 Regarding MOE**: Good point, this does bear some resemblance to MOE. In fact, it would be interesting future work to see if our strategy can be incorporated in model training as a principled approach to MOE. However, at the moment the two are different. MOE typically trains model parameters end-to-end for inference optimization, whereas P2L's approach predicts leaderboard coefficients over fixed sets of external, possibly black-box APIs. The P2L framework provides granular evaluation of external models without retraining these models or modifying their parameters.
>
> **W4 Method Clarity**: We thank the reviewer for pointing out clarity issues regarding terminology. In our revision, we will simplify and clearly define key terminology (e.g., BT regression, leaderboard vectors, routing policy notation) to improve readability, especially in Section 2.
>
> **W5 Comparison w/ Previous work**: We will make sure to provide more detail on our training method in future revisions. Our architecture is similar to reward modeling, with a pretrained transformer initialized with a new output linear layer. However, we output a dimension per model in the leaderboard, and use a loss that supervises 2 dimensions of the output per datapoint– this, to our knowledge, is novel.
>
> **W6 Baselines**: We understand the reviewers' concern around solid method baselines. We previously tested retrieval-based methods nearly exactly as described by the reviewer (e.g., retrieving k nearest prompts, weighting similar prompts). Empirically, we found the log-loss did not improve over marginal BT regression, with average loss differences smaller than 0.01 across various similarity metrics. Primarily, k must be quite large to obtain stable BT regression, which nullifies any per-prompt estimation advantage. Moreover, existing embeddings may encode “similar” in “semantic” subspace, and not “model performance” subspace. For example, “Explain BT regression” vs “Implement BT regression” are semantically similar, but fundamentally different tasks (explaining vs coding). Ultimately, we chose the marginal BT as baseline, since this was the most effective and stable baseline.
>
> ---
>
> We greatly appreciate the reviewer’s constructive feedback, which significantly enhances the quality and clarity of our work. We respectfully ask the reviewer to reconsider their rating, given our revisions, clarifications, and the substantial potential impact of our contributions to the community.

---

> > ### Comment · Reviewer_Hdbw · 2025-04-03
> >
> > Thanks the authors for the rebuttal.
> >
> > - **(Conditionally) Addressed**: W1, W3, W4. The authors promised better clarity and framing.
> > - **Partically Addressed**: W2, W6.
> > - **Not Addressed:** W5. I don't think the authors provide too much details.
> >
> > I will consider changing my score with more discussion with the other reviewers/ACs, given the authors addressed some of the concerns.

---

> > > ### Author Response · Authors · 2025-04-07
> > >
> > > We thank the reviewer for their valuable feedback and provide additional clarifications and revisions below.
> > >
> > > ---
> > >
> > > **W5:** Our approach employs a pre-trained transformer followed by a linear layer to estimate Bradley-Terry (BT) regression coefficients. We utilize a partially supervised BT loss, wherein two coefficients are supervised for each individual data point. Training leverages extensive pairwise human preference data. To our knowledge, this particular approach has not previously been explored. We agree with the reviewer that further detail on our methodology would strengthen the paper, and we will clarify the novelty and specifics of our training method in the revised manuscript. Additionally, we introduce methods for optimal routing and aggregating per-prompt BT coefficients, both of which, to our best knowledge, constitute novel contributions motivated by P2L.
> > >
> > > **W2:** We appreciate the reviewer highlighting concurrent work and recognize its relevance. However, in line with ICML reviewer guidelines on concurrent works (see "Concurrent Works,” https://icml.cc/Conferences/2025/ReviewerInstructions), authors are not expected to discuss research published after the submission deadline. Nevertheless, for clarity, we briefly highlight the distinction: LLMSelector is concerned with optimizing performance on a specific compound AI system marginally over a *task*. It is a framework that takes in a compound AI system, a training dataset, and some feedback signal (e.g. an LLM Judge), and returns an optimal configuration. On the other hand, P2L trains a model on a broader, non-task-specific set of human preferences. During inference, the P2L model provides a calibrated leaderboard of predicting LLM performance on any singular *prompt*— this process happens with virtually no latency, and without additional feedback data collection, such as LLM judgements. Thus, although both works involve model selection, their methodologies, objectives, and use-cases differ significantly.
> > >
> > > **W6:** We believe our approach and problem formulation offer meaningful novelty compared to existing work. Prior research primarily focuses on leaderboards based on aggregate metrics (e.g., average hit-rate or marginal correctness). In contrast, our method aims to extend evaluation to predict performance conditioned on individual prompts (i.e., E[correct | input] instead of E[correct]), enabled by extensive data and deep learning approaches. We suggest that the marginal leaderboard serves as an appropriate baseline for comparison, and our experiments demonstrate consistent improvements over this baseline. We hope this clarifies the intended contribution and significance of our work.
> > >
> > > ---
> > >
> > > We hope this clarifies the intended contribution and significance of our work. We thank the reviewer again for their thoughtful feedback and would appreciate their consideration in revising the score.

---

### Official Review · Reviewer_Lp4H · 2025-03-31

**Overall Recommendation:** 4

**Summary:**

This paper proposes a prompt-to-leaderboard (P2L) method to predict prompt-specific leaderboards via large language models (LLMs) trained on human preference data. The authors make LLMs to output the coefficients of parametric regressionsthat represent per-prompt leaderboards. Thus, this leaderboard supports optimal routing, personalized leaderboards, task-specific performance analysis, and automated evaluation of strengths and weaknesses. Empirical results show that P2L's router achieved the #1 spot in the Chatbot Arena leaderboard in January 2025.

### update after rebuttal

The authors' rebuttal solves my concerns. I hightly suggest the authors to incorporate the content in the rebuttal into the revision.

**Claims And Evidence:**

The claims made in the submission are supported by clear and convincing theoretical / empirical evidence.

**Essential References Not Discussed:**

Most of the essential related works are discussed.

**Experimental Designs Or Analyses:**

The experimental designs or analyses in Section 3 are mostly sound and valid.

**Methods And Evaluation Criteria:**

The proposed methods and corresponding evaluation criteria make sense for the research problem.

**Other Comments Or Suggestions:**

None.

**Other Strengths And Weaknesses:**

Strengths:

1. The proposed prompt-to-leaderboard method is interesting and novel, which provides meaningful insights into the nature of LLM leaderboards which are shaped by human preferences.
2. Well-designed experiments show the effectiveness of the proposed method from different perspectives, which are inspiring.
3. This paper is easy to follow.

Weaknesses:

1. I am curious about the generalization ability of P2L when new LLMs are incorporated into the leaderboard. This may be important because new LLMs are still emerging rapidly. How much preference data about new LLMs are enough for accurate leaderboard re-estimation when they added to the leaderboard? Can original P2L model parameters be utilized to accelerate the training process of P2L when new LLMs are incorporated? The authors are suggested to add more discussions about the generalization ability of P2L to new LLMs.

**Questions For Authors:**

I have included my questions in other parts of the review.

**Relation To Broader Scientific Literature:**

Compared with the broader scientific literature, this paper first (to the best of my knowledge) proposes a prompt-specific leaderboard generation method which can support optimal routing and task-specific performance analysis.

**Theoretical Claims:**

I have checked the correctness of most of the theoretical claims in this paper, especially Theorem 1.

---

> ### Author Rebuttal · Authors · 2025-04-01
>
> We thank the reviewer for their comments. We thank the reviewer for considering the novelty and soundness of our work. We agree with the reviewer on generalization to new LLMs; this is of interest, and we are excited to continue exploring methods, such as online learning, to optimize this in future work. Roughly, we have seen it takes roughly 6k votes to get stable rankings for a given model— similar to what is needed for the marginal case. Fast adaptation to new models is a promising future direction. We also note our P2L models are fairly inexpensive to train: P2L-7B on 1.5 million data points costs less than \\$250 to train end-to-end using a relatively unoptimized Deepspeed + Huggingface Trainer infrastructure (\\$23.92 per hour for 8xH100 on Runpod). The well-performing 3B and 1.5B variants train with negligible cost. We will include these exact training hardware, time, and cost numbers in the appendix for future revisions.

---

### Decision · Program_Chairs · 2025-05-01

**Decision:**

Accept (poster)

**Comment:**

This paper introduces Prompt-to-Leaderboard (P2L), a method for generating prompt-specific leaderboards of large language models (LLMs) by training LLMs on human preference data. The core idea is to model prompt-dependent Bradley-Terry coefficients, enabling per-prompt performance comparisons. Key applications include optimal query routing, personalized evaluations, and automated model strength/weakness analysis. The authors validate P2L on Chatbot Arena and LiveBench, demonstrating that P2L routers outperform static models in live evaluations and generalize well to out-of-distribution tasks. Scaling experiments suggest P2L follows power-law improvements with model size and data.

The scores, 4, 2, 3, 3, tend towards accept, and some scores were raised after rebuttal, but based on promised improvements and materials release by the authors.

Overall I am recommending weak accept.

Strengths:

In general the reviewers assessed that the paper's contributions are sound and would be of general interest.  The paper combines parametric statistics with LLMs for prompt-adaptive evaluation, which is novel and impactful. The power-law trend (Section 3.2) is convincing.
The method is interesting and novel, which provides meaningful insights into the nature of LLM leaderboards which are shaped by human preferences.  The experiments are well-designed and show the effectiveness of the proposed method from different perspectives, which are inspiring.

Weaknesses:

Several author commented on a lack of clarity in the writing of the paper:

“I have some concerns about the clarity of the presentation and the rigor of the conclusions which I would like to see addressed. The authors overclaim in several places, which is unnecessary, as the work is sufficiently interesting without hyperbole.” And “The method is clear, but the writing is a bit hard to follow”

Other weaknesses noted are: the method is not scalable enough, the generalization ability of P2L when new LLMs are incorporated into the leaderboard is questioned.  The method seems to incur additional computational overhear that may not justify the performance gains.

The reviewers also commented that although LiveBench and Chatbot Arena are standard benchmarks the derived category-specific leaderboards (Section 3.4) rely on automated clustering without human validation that may result in noisy or subjective categories, and lead to ambiguous categorization criteria.

The  simulated costs (Section 3.3.2) rely on token-length averages, ignoring variance in real deployments.